# Novel Insights of T2-Weighted Imaging: Significance for Discriminating Lung Cancer from Benign Pulmonary Nodules and Masses

**DOI:** 10.3390/cancers13153713

**Published:** 2021-07-23

**Authors:** Katsuo Usuda, Shun Iwai, Aika Yamagata, Yoshihito Iijima, Nozomu Motono, Munetaka Matoba, Mariko Doai, Keiya Hirata, Hidetaka Uramoto

**Affiliations:** 1Department of Thoracic Surgery, Kanazawa Medical University, Ishikawa 920-0293, Japan; mhg1214@kanazawa-med.ac.jp (S.I.); aicarby@kanazawa-med.ac.jp (A.Y.); y-iijima@kanazawa-med.ac.jp (Y.I.); motono@kanazawa-med.ac.jp (N.M.); hidetaka@kanazawa-med.ac.jp (H.U.); 2Shimada Hospital, Fukui 910-0855, Japan; 3Department of Radiology, Kanazawa Medical University, Ishikawa 920-0293, Japan; m-matoba@kanazawa-med.ac.jp (M.M.); doaimari@kanazawa-med.ac.jp (M.D.); 4MRI Center, Kanazawa Medical University Hospital, Ishikawa 920-0293, Japan; keiya@kanazawa-med.ac.jp

**Keywords:** T2-weighted imaging (T2WI), magnetic resonance imaging (MRI), lung cancer, pulmonary nodule and mass (PNM), pulmonary abscess

## Abstract

**Simple Summary:**

The aim of this research was to clarify whether T2WI is efficient for discriminating lung cancer from BPNMs. A T2 contrast ratio (T2 CR) for a pulmonary nodule is defined as the ratio of T2 signal intensity of a pulmonary nodules divided by the T2 signal intensity of the rhomboid muscle. There were 52 lung cancers and 47 BPNMs. The optical cutoff value for malignancy was 2.44 for T2 CR by receiver operating characteristic curve. The T2 CR (2.14 ± 0.63) of lung cancers was significantly lower than that (2.68 ± 1.04) of BPNMs (*p* = 0.0021). The T2 CR of lung cancers was significantly lower than that (2.93 ± 0.26) of pulmonary abscesses (*p* = 0.011). T2 CR is efficient in discriminating lung cancer from BPNMs.

**Abstract:**

Diffusion-weighted imaging is useful for discriminating lung cancer from benign pulmonary nodules and masses (BPNMs), however the diagnostic capability is not perfect. The aim of this research was to clarify whether T2-weighted imaging (T2WI) is efficient in discriminating lung cancer from BPNMs, especially from pulmonary abscesses. A T2 contrast ratio (T2 CR) for a pulmonary nodule is defined as the ratio of T2 signal intensity of a pulmonary nodule divided by the T2 signal intensity of the rhomboid muscle. There were 52 lung cancers and 40 inflammatory BPNMs (mycobacteria disease 12, pneumonia 13, pulmonary abscess 9, other 6) and seven non-inflammatory BPNMs. The T2 CR (2.14 ± 0.63) of lung cancers was significantly lower than that (2.68 ± 1.04) of BPNMs (*p* = 0.0021). The T2 CR of lung cancers was significantly lower than that (2.93 ± 0.26) of pulmonary abscesses (*p* = 0.011). When the optical cutoff value of T2 CR was set as 2.44, the sensitivity was 0.827 (43/52), the specificity 0.596 (28/47), the accuracy 0.717 (71/99), the positive predictive value 0.694 (43/62), and the negative predictive value 0.757 (28/37). T2 CR of T2WI is useful in discriminating lung cancer from BPNMs. Pulmonary abscesses, which show strong restricted diffusion in DWI, can be differentiated from lung cancers using T2WI.

## 1. Introduction

Pulmonary nodules and masses (PNMs) are abnormal findings on chest radiographies. Among PNMs, lung cancer is the most critical disease related to deaths, and its correct diagnosis is essential for patients. Today, patients with PNMs detected on radiographs undergo early CT scans. It has been highlighted in several recent publications on CT screening for lung cancer that the majority of smokers who undergo thin-section CT have been found to have small lung nodules, most of which were smaller than 7 mm in diameter, and that the vast majority were benign and were different from larger lung nodules in chest radiographs [1]. The radiographic features accepted as a benign nodule are a stable nodule size for a two year period or longer, presence of fat in the nodule, and characteristic concentric, central, or stippled calcification patterns [2]. For contrast-enhanced CT, PNMs that can be enhanced by more than 20 Hounsfield units (HU) after the administration of intravenous contrast medium are usually malignant, whereas PNMs that can be enhanced less than 15 HU are benign [3]. A recent meta-analysis of ten contrast-enhanced CT studies showed a pooled sensitivity of 93%, a specificity of 76%, a positive predictive value (PPV) of 80%, and a negative predictive value (NPV) of 95% for PNMs [4].

18-fluoro-2-deoxy-glucose positron emission tomography/computed tomography (FDG-PET/CT) is available for discriminating malignant from benign pulmonary nodules [5,6]. Concerning the imaging modality of PNMs, FDG-PET/CT are utilized widely. Its maximum standardized uptake value (SUVmax) expresses glucose uptake and represents how invasive the tumor is. However, FDG-PET/CT’s shortcomings include false-negative results for well-differentiated pulmonary adenocarcinomas [7], metabolically active tumors of small volume [8], and false-positive results for inflammatory benign nodules [9]. The limitations of FDG-PET/CT are the availability and high costs [10]. Patients with PNMs are usually recommended to undergo one of the available invasive procedures (such as bronchoscopy, transthoracic needle biopsy, thoracoscopic surgery) for obtaining specific tissue diagnosis. Most of the currently available non-invasive procedures cannot clearly differentiate between benign and malignant PNMs.

For the last three decades, magnetic resonance imaging (MRI) for lung cancer has been utilized sparingly in the lung cancer cases of mediastinum invasion or chest wall invasion, after Webb et al. of the Radiologic Diagnostic Oncology Group [11] recommended limited usage of MRI in 1991. While CT and FDG-PET/CT are currently the most commonly clinically employed modalities for pulmonary nodule staging, the augmentative potential of MRI has been presented [12]. At present, lung MRI makes it possible to replace up to 90% of CT examinations with radiation-free magnetic resonance diagnostics of the lungs, without suffering any diagnostic loss [13]. This applies, in particular, to children, who repeatedly require sectional imaging of the lung [13]. MRI yielded high sensitivity for the detection of pulmonary nodules and enabled accurate assessment of their diameter [14]. Therefore, it may be considered an alternative to CT for follow-up of some lung lesions [14]. Diffusion-weighted magnetic resonance imaging (DWI) is able to detect the decreased diffusion of water molecules. Its apparent diffusion coefficient (ADC) value shows a quantitative value for the diffusion of water molecules in biological tissues, and the ADC of malignancy is significantly lower than that of normal organs or benign tumors [15]. Although all of the meta-analyses confirmed that DWI could discriminate malignancy from benignity for PNMs [16,17,18], it was difficult to discriminate a pulmonary abscess from a lung cancer, because a pulmonary abscess shows strong restricted diffusion. T2-weighted imaging (T2WI) is an essential MRI imaging technique. Its efficacy has been demonstrated, especially in the assessment of high-intensity fluid in lesions, such as edema, cysts, mucus, intratumoral necrosis, and hemorrhage [19,20]. A cystic mediastinal tumor presents a high signal intensity, showing fluid inside on a T2WI [21]. On the other hand, lung cancers are likely to express a lower intensity compared to BPNMs in T2WI. T2WI can show a qualitative evaluation of the water of a lesion.

The aim of this study was to clarify whether T2WI is efficient in discriminating lung cancer from BPNMs, especially from pulmonary abscess, and to determine the qualitative evaluation of the characteristics of the water of T2WI for lung cancers and BPNMs.

## 2. Materials and Methods

### 2.1. Eligibility

The research protocol for assessing MRI in patients with PNMs was approved by the ethical review board of Kanazawa Medical University (the approval number: No. I302). A written informed consent for MRI was obtained from each patient after discussing the risks and benefits of the examinations.

### 2.2. Patients

This is a retrospective study. Of the patients who had primary lung cancers or benign pulmonary nodules and masses (BPNMs), and had MRI examinations before pathological diagnosis or bacterial diagnosis from May 2009 to April 2018, 99 patients who qualified for detailed analysis of MRI were enrolled in this research (Table 1). Patients included in the research had PNMs with a maximum size of 10 cm or less, and which possessed no definitive calcification. Most of the PNMs were pathologically diagnosed by resection or through bronchoscopy. The other remaining PNMs had a diagnosis by bacterial culture or roentgenographically follow-up study. PNMs were diagnosed as benign when the PNMs decreased in size or disappeared upon review of retrospective x-ray films or CT. Pure ground-glass-nodule (GGN)-type lung cancers were excluded from the research. No patients had received prior treatment. Sixty-six patients were men and 33 were women. The mean age was 67 years (range 43–85). There were 52 lung cancers and 47 BPNMs. The diagnosis was made pathologically in 52 lung cancers and 31 BPNMs, and using a bacterial culture in 3 BPNMs. The remaining 13 BPNMs were diagnosed as pneumonia by decreased size or disappearance of the BPNMs. Of the 52 lung cancers, there were 33 adenocarcinomas, 16 squamous cell carcinomas, 1 large cell neuroendocrine carcinoma (LCNEC), 1 large cell carcinoma, and 1 small cell carcinoma. There were 4 pathological T1a (pT1a) carcinomas, 5 pT1b carcinomas, 17 pT2a carcinomas, 12 pT2b carcinomas, 13 pT3 carcinomas, and 1 pT4 carcinoma. There were 32 pathological pN0 (pN0) carcinomas, 13 pN1 carcinomas, and 7 pN2 carcinomas. There were 49 M0 carcinomas and 3 M1a carcinomas. There were 9 pStage IA, 13 pStage IB, 3 pStageIIA, 6 pStage IIB, 12 pStage IIIA, and 3 pStage IV. The new definitions in UICC 8 [22] were used for TNM classification and the lymph node stations of lung cancer. For 47 BPNMs, there were 40 inflammatory BPNMs (mycobacteria disease 12 (tuberculosis 5, nontuberculous mycobacteria 7), pneumonia 13, pulmonary abscess 9, organized pneumonia 2, pulmonary scar 2, and pulmonary granuloma 2, and 7 non-inflammatory BPNMs (hamartoma 3, pulmonary sequestration 1, nodular lymphoid hyperplasia 1, inflammatory pseudotumor 1, and encapsulated pleural effusion 1)).

### 2.3. MR Imaging

All MR images were made with a magnetic scanner of 1.5 T (Magnetom Avanto; Siemens, Erlangen, Germany). The conventional MR images were a coronal T1-weighted spin-echo sequence and coronal and axial T2-weighted fast spin-echo. Examinations of the 1.5-T MRI were performed as follows: T2WI was obtained in a turbo spin echo (TSE); (TR/TE, 4400–6000/74 ms; FOV, 350 × 240 mm; matrix, 320 × 198; thickness, 6.0 mm), Flip angle 90°. T1-weighted imaging (T1WI) was obtained in gradient recalled echo (GRE) (VIBE; TR/TE, 6.54/4.78 ms; FOV, 380 × 240 mm; matrix, 256 × 151; thickness, 3.5 mm). Regions of interest (ROIs) were selected on each pulmonary nodule detected on each of the sequences and on the rhomboid muscle. Based on the definition of Koyama et al. [23], T2 contrast ratio (T2 CR) was defined as the ratio of T2 signal intensity of a pulmonary nodule divided by the T2 signal intensity of the rhomboid muscle. The ROIs on the muscle were depicted at 120 mm^2^ (a circle 8 mm in size). T2 signal intensities of PNMs were obtained by drawing round, elliptical, or free-hand ROIs on lesions, which were detected visually on the T2WI. A radiologist (M.D.) with 25 years of MRI experience, who was not aware of the patients’ clinical data, and a pulmonologist (K.U.) with 28 years of experience assessed the MRI data. All measurements were performed by one experienced author (K.U.) supported by the experienced radiologist (M.D.). There were no discrepancies in the data between them.

### 2.4. Statistical Analysis

The data are expressed as the mean ± standard deviation. A two-tailed Student’s t-test was used for comparison of the T2 CR values of two groups. A receiver operating characteristic (ROC) curve was applied to assess the diagnostic efficacy of T2 CR values in terms of malignant–benign differentiation. The statistical analyses were made with StatView for Windows (Version 5.0; SAS Institute Inc., Cary, NC, USA). A *p* value of <0.05 was defined as statistically significant.

## 3. Results

Adenocarcinoma (case 1), squamous cell carcinoma (case 2), non-tuberculous mycobacteria (case 3), and hamartoma (case 4) were presented in the CT lung window setting (a), CT mediastinal window setting (b), and T2 WI (c) (Figure 1). T2 CRs were 2.03 (case 1) (true positive), 2.43 (case 2) (true positive), 3.52 (case 3) (true negative), and 2.95 (case 4) (true negative).

In the ROC curves of T2 CR for all the PNMs (Figure 2), the area under the ROC curve (AUC) was 64.8% and the 95% confidence interval was 54.0% to 76.4%. When the cutoff value of T2 CR was set as 2.44, the sensitivity was 0.827 (43/52), the specificity was 0.596 (28/47), the accuracy was 0.717 (71/99), the PPV was 0.694 (43/62), and the NPV was 0.757 (28/37).

The T2 CR (2.14 ± 0.63) of lung cancers was significantly lower than that (2.68 ± 1.04) of BPNMs (*p* = 0.0021) (Figure 3). The T2 CR of lung cancers was significantly lower than that (2.53 ± 0.87) of inflammatory BPNMs (*p* = 0.017) and significantly lower than that (3.36 ± 1.73) of non-inflammatory BPNMs (*p* = 0.0005) (Figure 4). The T2 CR of inflammatory BPNMs was not significantly lower than that of non-inflammatory BPNMs (*p* = 0.057).

T2 WIs are presented based on each diagnosis of PNMs (Figure 5). The T2 CR of lung cancers was significantly lower than that (2.93 ± 1.26) of pulmonary abscesses (*p* = 0.0069), and not significantly lower than that (2.49 ± 0.80) of mycobacteria infections (*p* = 0.11).

For T2 CR based on pT factors and pN factors, there were not any significant differences among pT factors or pN factors (Figure 6). For T2 CR based on pM factors and pStages, there were not any significant differences among pM factors or pStages (Figure 7).

## 4. Discussion

Koyama et al. [23] reported that non-contrast enhanced pulmonary MRIs can effectively confirm malignant nodules as well as a thin-section multidetector CT (MDCT). MRI can identify and stage lung cancer, and this method could be an outstanding alternative to CT or PET/CT for the assessment of pulmonary malignancies and other diseases [24]. Conventional MR sequences can show the essential differences between mass-like tuberculosis and lung cancer and may be useful for differentiating pulmonary masses [25]. MRI is more useful than CT for the visualization of the heart, the pericardium, and mediastinal vessels [26]. MRI can have an advantage for specifically investigating invasion of the superior vena cava or myocardium, or extension of the tumor into the left atrium via the pulmonary veins [26]. Although FDG-PET/CT is thought to be more appropriate for this purpose, MRI’s strong point is its universal availability and being less expensive [24].

DWI was described as able to discriminate malignancy from benignity for PNMs [16,17,18], and a pulmonary abscess showed strong restricted diffusion in DWI. Some pathologic diseases, such as pulmonary abscess, chronic pneumonia, pulmonary tuberculosis, nontuberculous mycobacteria, sarcoidosis, scars, and other inflammatory or infectious conditions behave like malignant diseases by exhibiting restricted diffusion [27,28]. On the other hand, carcinomas would show diffusion restriction because of cells with a large sized nucleus, high nucleus-cytoplasm rate, intracellular macromolecules, the limited size of the extracellular matrix, and high cellular proliferation rate [29,30].

We focused on the strengths of T2WI. Traditional T2WI can detect effusion in the body or in the tumor as a high intensity signal (showing bright). Pleural effusion and cystic mediastinal tumors are easy to diagnose with T2WI. T2WI expresses fluid in the lesion brightly and is suitable for detection of pulmonary tuberculosis, nontuberculous mycobacteria, pulmonary abscess, chronic pneumonia, and other infectious or inflammatory conditions. In particular, T2WI may help tell the difference between lung cancer and progressive massive fibrosis (PMF) [31]. Mean T2 signal intensity ratios also differed significantly between benign and malignant lesions [32]. The signal intensity ratios (SIRs) of the lesion divided by the rhomboid muscle on T2WI and T1WI were significantly different between mass-like tuberculosis and lung cancer [25]. T2-weighted signal intensity (SI) ratio (SI_nodule_/SI_psoas muscle_) can differentiate metastases from lipid-poor adenomas [33].

Using a ROC curve of T2 CR for all the PNMs, the area under the ROC curve was 64.8%, which was moderate, and not a high value. When the optical cutoff value of T2 CR was set as 2.44, the sensitivity was 0.827, the specificity 0.596, and the accuracy 0.717, which were also not very high. The T2 CR (2.19 ± 0.56) of the lung cancers was significantly lower than that (2.62 ± 1.08) of BPNMs. The T2 CR (2.19 ± 0.56) of lung cancers was significantly lower than that (2.50 ± 0.87 × 10^−3^ mm^2^/s) of inflammatory BPNMs (*p* = 0.039) and significantly lower than that (3.45 ± 1.71) of non-inflammatory BPNMs. The more a PNM can present the characteristics of water, the higher T2 CR value it can have, and it can show a brighter image in T2 WI. This evidence could indicate that BPNMs had greater qualitative characteristics of water than lung cancer. T2 CR of T2WI is useful for differential diagnosis of PNMs and for the qualitative evaluation of the characteristics of the water of PNMs.

In the literature, there have been several articles concerning the diagnostic performance of T2WI and DWI for differential diagnosis for many other organs of the body. T2WI combined with DWI may be a useful tool for detecting prostate cancer in the overall evaluation of prostate cancer [34,35], and myometrial invasion and staging of endometrial carcinoma [36,37]. Diagnostic possibilities would be increased after T2WI and DWI are fused for the diagnosing of lung cancer and BPNMs [38]. A previous paper [38] dealt with a whole MRI value of DWI and T2WI and determined the effectiveness of, not only DWI, but also T2WI in the assessment of the differential value and a combined assessment of how effective they were for discriminating PNMs. On the other hand, this paper only dealt with the diagnostic efficacy of T2WI for discriminating PNMs. To date, the diagnostic efficacy of T2WI was not clear for discriminating PNMs. T2 CR of T2WI is useful in discriminating lung cancer from BPNMs and in qualitative evaluation of the water content in PNMs. It is very important that a pulmonary abscess which shows strong restricted diffusion in DWI can be differentiated from lung cancers using T2WI.

MRI involves no radiation exposure, as well as no contrast mediums, and is ideal for the examination of pregnant women and children. In the next decade, MRI will become more available for PNM assessment, because CT and FDG-PET/CT have some risks of radiation exposure. Not only do we have to inform our patients of the risks of radiation exposure and dangers, we also must get informed consent that our patients understand the risks and agree with the procedure. This can be unexpected for some patients, causing undue worry.

We have to keep in mind that this research had two limitations. First, it was a retrospective study and was conducted at a single institution. The research dealt with a small number of patients. Further research is necessary to assess the performance of T2 WI for differentiating between lung cancers and BPNMs.

## 5. Conclusions

When the optical cutoff value of T2 CR was set as 2.44, the sensitivity was 0.827, the specificity 0.596, and the accuracy 0.717. T2 CR of lung cancers was significantly lower than that of BPNMs (*p* = 0.013). T2 CR of lung cancers was significantly lower than that of inflammatory BPNMs (*p* = 0.039) and significantly lower than that of non-inflammatory BPNMs (*p* = 0.0001). T2 CR of lung cancer was significantly lower than that of pulmonary abscesses, which shows strong restricted diffusion in DWI can be differentiated from lung cancers using T2WI. T2 CR is not correlated to TNM classification of lung cancer. T2 CR of T2WI is useful for the differential diagnosis of PNMs and for the qualitative evaluation of the characteristics of the water of PNMs.

## Figures and Tables

**Figure 1 cancers-13-03713-f001:**
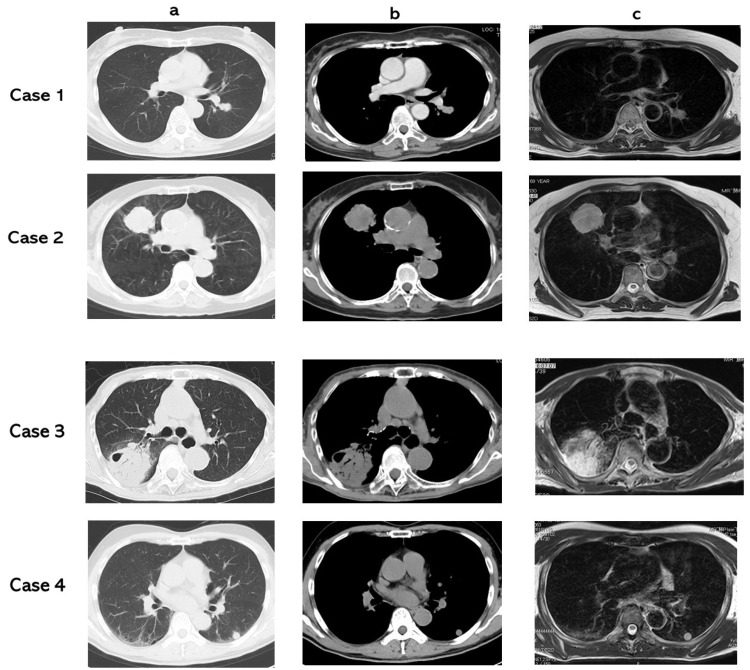
(**a**): CT lung window setting, (**b**): CT mediastinal window setting, (**c**): T2 WI. Case 1: adenocarcinoma, T2 CR: 2.03. Case 2: squamous cell carcinoma, T2 CR: 2.43. Case 3: non-tuberculous mycobacteria, T2 CR: 3.52. Case 4: hamartoma, T2 CR: 2.95.

**Figure 2 cancers-13-03713-f002:**
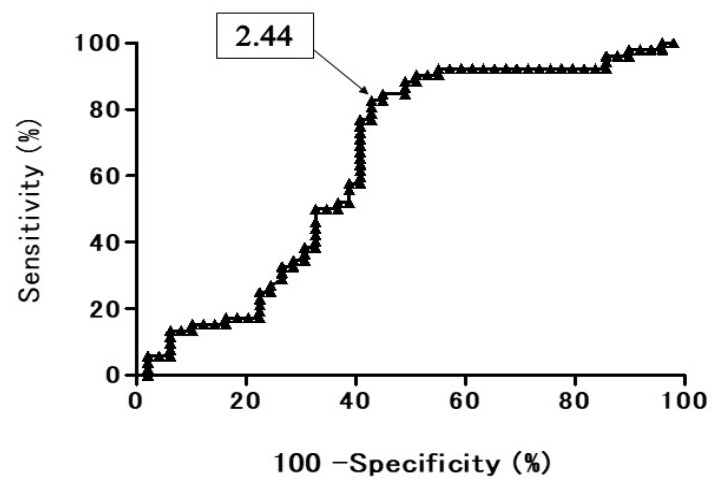
Receiver operating characteristic (ROC) curve presents the diagnostic performance of T2 CR for discriminating benign pulmonary nodule and mass (BPNM) from lung cancer. Area under the ROC curve (AUC) was 64.8% and the 95% confidence interval was 54.0% to 76.4%. T2 CR = 2.44, sensitivity 0.827 (43/52), specificity 0.596 (28/47), accuracy 0.717 (71/99), PPV 0.694 (43/62), and NPV 0.757 (28/37).

**Figure 3 cancers-13-03713-f003:**
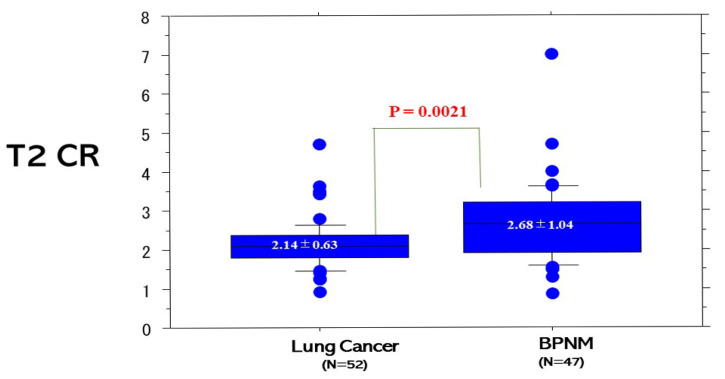
T2 contrast ratio (T2 CR) between lung cancer and BPNM. T2 CR (2.14 ± 0.63) of lung cancer was significantly lower than that (2.68 ± 1.04) of BPNMs (*p* = 0.0021).

**Figure 4 cancers-13-03713-f004:**
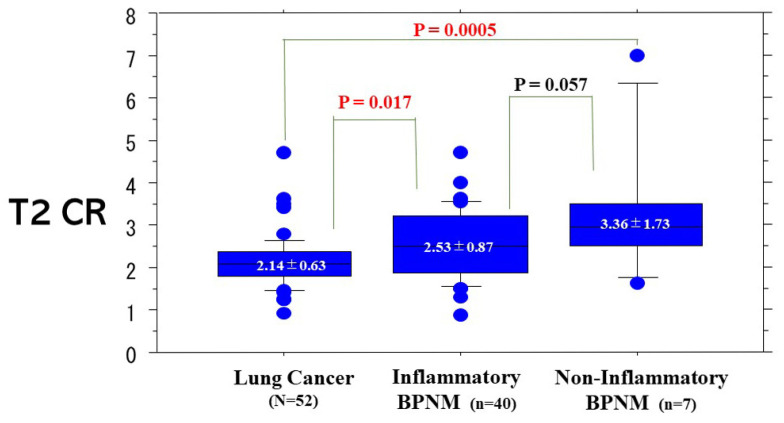
T2 Contrast ratio (CR) among lung cancers, inflammatory BPNMs and non-inflammatory MPNMs. T2 CR (2.14 ± 0.63) of lung cancers was significantly lower than that (2.53 ± 0.87) of inflammatory BPNMs (*p* = 0.011) and significantly lower than that (3.36 ± 1.73) of non-inflammatory BPNMs (*p* = 0.0005).

**Figure 5 cancers-13-03713-f005:**
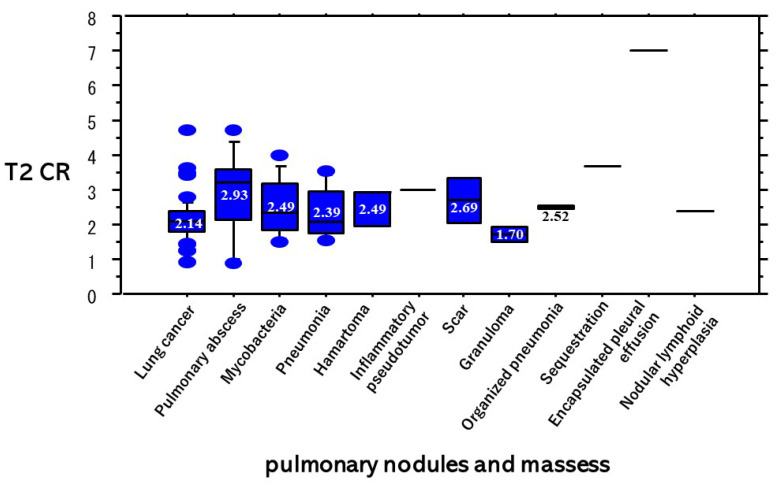
T2 WIs based on each diagnosis of PNMs. The T2 CR (2.14 ± 0.63) of lung cancer was significantly lower than that (2.93 ± 1.26) of pulmonary abscesses (*p* = 0.0069) and not significantly lower than that (2.49 ± 0.80) of mycobacteria infections (*p* = 0.11).

**Figure 6 cancers-13-03713-f006:**
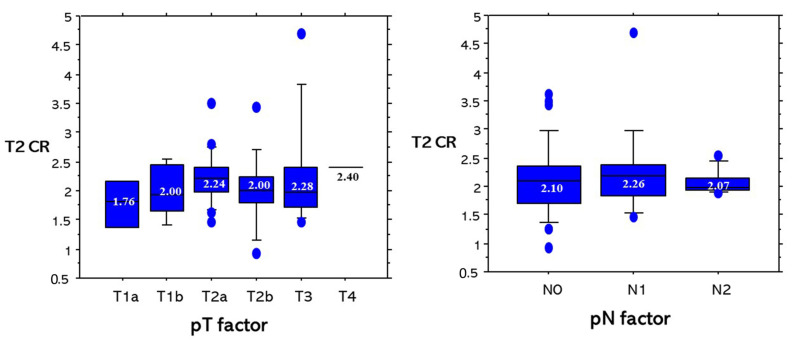
T2 CR based on pT factors and pN factors. There were no significant differences among T factors or N factors.

**Figure 7 cancers-13-03713-f007:**
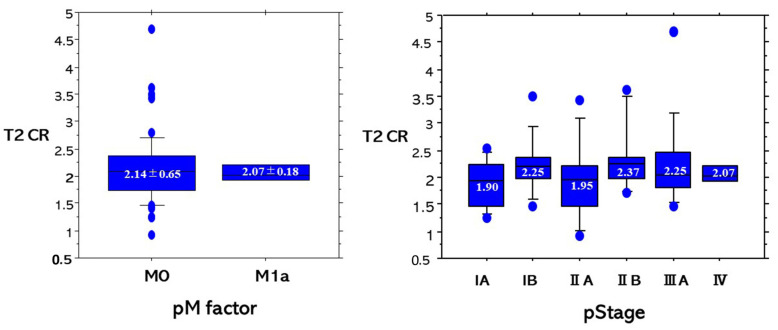
T2 CR based on pM factors and pStages. There were no significant differences among pM factors or pStages.

**Table 1 cancers-13-03713-t001:** Patient characteristics.

Diagnosis		No. of Patients	
Lung cancer		52	
	Adenoca.		33
	Squamous cell ca.		16
	LCNEC		1
	Large cell ca.		1
	Small cell ca.		1
Inflammatory benignity		40	
	Mycobacteria disease		12 (Tbc 5, NTM 7)
	Pneumonia		13
	Pulmonary abscess		9
	Organized pneumonia		2
	Pulmonary scar		2
	Pulmonary granuloma		2
Non-inflammatory benignity		7	
	Hamartoma		3
	Pulmonary sequestration		1
	Nodular lymphoid hyperplasia		1
	Inflammatory pseudotumor		1
	Encapsulated pleural effusion		1
		99	

## Data Availability

The data presented in this study are available in this article.

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
