# Peer review of "Novel Insights of T2-Weighted Imaging: Significance for Discriminating Lung Cancer from Benign Pulmonary Nodules and Masses"

_cancers, 2021, doi:10.3390/cancers13153713_

Round 1

Reviewer 1 Report

Dear authors,

the manuscript has improved significantly and it now suitable for publication.

Author Response

Answers to Reviewer 1’s Comments and Suggestions for Authors

Dear authors,

the manuscript has improved significantly and it now suitable for publication.

I really appreciated the reviewer’s comments.  With their feedback I was able to improve my paper.

Reviewer 2 Report

The manuscript entitled "Novel Insights of T2-weighted imaging: Significance for Discriminating Lung Cancer from Benign Pulmonary Nodules and Masses" highlgihted that T2 CR is efficient in discriminating lung cancer from BPNMs.

Minor comments:

  • The Authors should provide the expand forms for alla acronyms through the text when they first appear.
  • Minor spell check is required.

Author Response

Answers to Reviewer 2’s Comments and Suggestions for Authors

The manuscript entitled "Novel Insights of T2-weighted imaging: Significance for Discriminating Lung Cancer from Benign Pulmonary Nodules and Masses" highlgihted that T2 CR is efficient in discriminating lung cancer from BPNMs.

Thank you for your considerations.

Minor comments:

The Authors should provide the expand forms for all acronyms through the text when they first appear.

Minor spell check is required.

Thank you for your advice.  Mr. Dustin Keeling, who is a native English speaker, has checked this paper again to correct for misspellings and the full acronyms will be provided when first mentioned throughout the text.  Due to space constraints acronyms only will be used in the simple summary.

This manuscript is a resubmission of an earlier submission. The following is a list of the peer review reports and author responses from that submission.

Round 1

Reviewer 1 Report

Concern

The authors recently published a similar paper in Cancers "Cancers 2021, 13, 1551. https://doi.org/10.3390/cancers13071551" based on similar measurement technique but slightly different analysis strategy. Comparing Figure 2 in Cancers 2021, 13, 1551. https://doi.org/10.3390/cancers13071551
and figure 2 in current manuscript, the area under the curve is smaller in the new manuscript for the same type of analysis and comparison. This raise the concern with regard to the added value of the new study. The authors should discuss their previous manuscript already in the introduction and clearly explain what is the added value in the new study compared to the previous. Currently, the previous publication is mentioned in a single sentence in the discussion. This is actually not a discussion of current and past result, but rather a weak statement about a result in previous publication. Furthermore, the authors must access if there is a significant difference in classification performance. Making the image data with clinical annotation publically available could also increase the interest in the manuscript.

Minor:
Abbreviations are normally written out in abstract, summary and on first occurrence in main text.
1. Improve in abstract "T2 CR of T2WI" 
2. AU:  "However, FDG-PET/CT is easy to express.." -> "However, FDG-PET/CT shortcomings include..."
3. This claim "Actually, T2 WI is reliable for the diagnosis of cystic mediastinal tumors." needs reference
4. Figure 2 better to insert 95% confidence interval
5. Figure 1. The legend should be a readable text. a) and b) are both labeled CT - it is better to state what makes (a) and (b) different.
6. The first paragraph is not readable main text, but rather a table list. The summary of T2 CR can be written more concise.
7. AU: "can effectively find out malignant nodules " -> "can effectively identify..."